# The FC Algorithm to Estimate the Manning's Roughness Coefficients of Irrigation Canals

**Enrique Bonet [1],\*, Beniamino Russo [2] , Ricard González [2], Maria Teresa Yubero [3] , Manuel Gómez [2]
and Martí Sánchez-Juny [2]**

[1] Water Technology Center (CETAQUA), Ctra. d'Esplugues, 75, 08940 Cornella de Llobregat, Spain
[2] Flumen Institute, Barcelona School of Civil Engineering, C/Jordi Girona, 1, 08034 Barcelona, Spain; beniamino.russo@upc.edu (B.R.); ricardgonzalezblanch@gmail.com (R.G.); manuel.gomez@upc.edu (M.G.); marti.sanchez@upc.edu (M.S.-J.)
[3] Department of Civil and Environmental Engineering, Barcelona School of Civil Engineering, C/Jordi Girona, 1, 08034 Barcelona, Spain; maria.teresa.yubero@upc.edu
\* Correspondence: enrique.bonet@upc.edu or enric.bonet@cetaqua.com

**Abstract:** Freshwater scarcity has driven the integration of technological advancements and automation systems in agriculture in order to attempt to improve water-use efficiency. For irrigation canals, water-use efficiency is, in great measure, limited by the performance of management systems responsible for controlling the flow and delivering water to the farmers. Recent studies show a significant sensitivity of the results obtained from irrigation canal control algorithms with respect to the Manning's roughness coefficient value, thus, highlighting the importance of its correct estimation to ensure an accurate and efficient water delivery service. This is the reason why the friction coefficient algorithm was developed, to monitor the real behaviour of any irrigation canal by calculating the Manning's roughness coefficient constantly. The friction coefficient algorithm was conceived as a powerful offline tool that is integrated in a control diagram of any irrigation canal, concretely in an optimization control algorithm, which can reconfigure canal gates according to the current crop water demand and the real Manning's roughness coefficient values. The friction coefficient algorithm has been applied in several irrigation canals and different scenarios, with accurate results obtaining an average Manning coefficient deviation among $2 \times 10^{-4}$ and $4.5 \times 10^{-4}$.

**Keywords:** agricultural demands; irrigation canal control; Manning's roughness coefficient; parameter identification; open channel flow; optimization algorithms





## 1. Introduction

Though water covers approximately 71% of Earth's surface, only a small percentage (2.5% of the total water) is deemed profitable by the great majority of living organisms, which is generally referred to as freshwater. In fact, only 1.2% of fresh water (0.03% of total water on Earth) is considered to be easily accessible surface water and is, for the most part, found in ground ice, lakes, rivers, swamps, and the atmosphere, while the remaining 98.8% of freshwater is either located in glaciers and ice caps or composed of groundwater (see, Figure 1). The scarcity of reachable freshwater may be further aggravated by the current prospects of future human water consumption. The world population is rapidly growing, with expectations of reaching 8 billion people in 2030 and just shy of 10 billion people in 2050 according to United Nations [1], both of which are figures that suggest an imminent drive in global food demand. More precisely, recent studies place estimations of food consumption at an increase of 50% by 2030 and 70% by 2050 as a result of population growth, higher income per capita, and structural changes in diets, according to Alexandros and Bruinsma [2].

Bearing in mind the fact that agriculture is one of the main sources for providing food to the population and that 70% of the global freshwater is destined for irrigated agriculture

purposes according to United Nations [3], it becomes evident that freshwater demand will inevitably escalate to an unprecedented level.

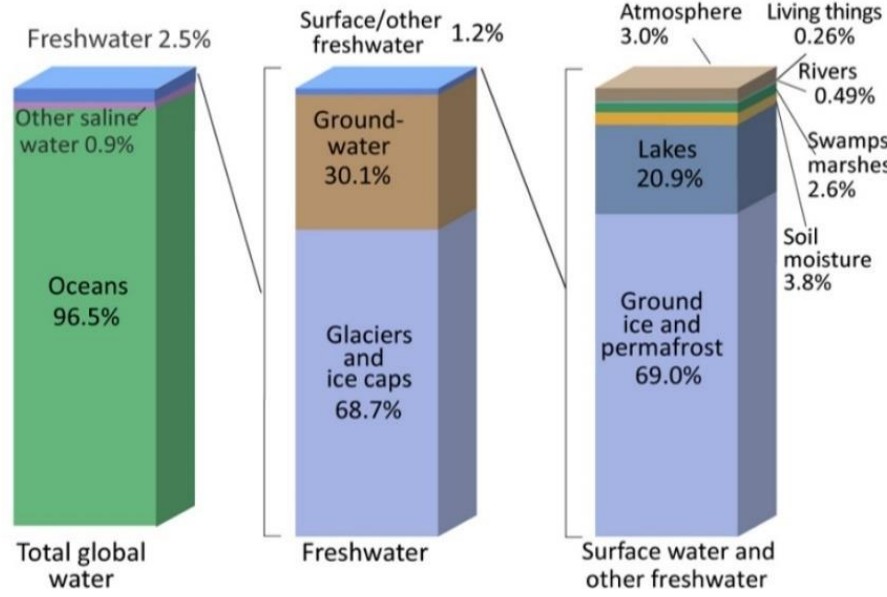

**Figure 1.** Water distribution on Earth [4].

This is the reason why governments realized that they must face freshwater scarcity by driving the integration of technological advancements and automation systems in agriculture in order to attempt to improve water-use efficiency [5]. In that sense, control algorithms aid us to increase the efficiency in canal management [6,7]. Many algorithms, in particular open-loop algorithms, have difficulties and failures in the implementation in real canals [8]. These difficulties are due to deviations between the predictive/control model and the reality frequently because disturbances are quite difficult to be considered.

The disturbances introduced into the canal lead to important deviations between the model and the reality. In that sense, Manning's roughness coefficient values are behind the deviations between models and reality, in the case of these values not being carefully identified. For instance, Manning's roughness coefficient does not change suddenly but progressively, introducing an important accumulated water level error along the canal.

The main problem in an irrigation canal is the disturbances caused by climatic variations (rainfalls and associated runoff), unscheduled demands by farmers (due to soil moisture and crop water requirements), and Manning roughness coefficient value errors, which are more difficult to mitigate by a controller. In such a case, an overall control diagram was proposed regarding the CSE algorithm [9] (see Figure 2), which is an excellent tool to approach the unscheduled demands in a canal (in real time), and the friction coefficient algorithm (FC algorithm), proposed in this paper, is a useful tool to identify the real Manning roughness coefficients in a canal (off-line), which is actually the research gap, and several authors have introduced different approaches to solve the issue [10–12].

In case such disturbances are identified, feedforward controllers such as GoRoSo [13,14] and/or feedback controllers such as GoRoSoBo [15], and predictive control [16], could provide canal gate trajectories to reach the water management objective, that is, keep the water level at cross-sections at the target water level.

In this paper, the FC algorithm has been described and tested in several practical examples. First at all, the algorithm was tested in a canal with two single pools; the geometry of the canal is based on previous works [9]. In a second example, the algorithm was also tested with the tests cases [17] introduced by the ASCE task committee on canal automation algorithms.

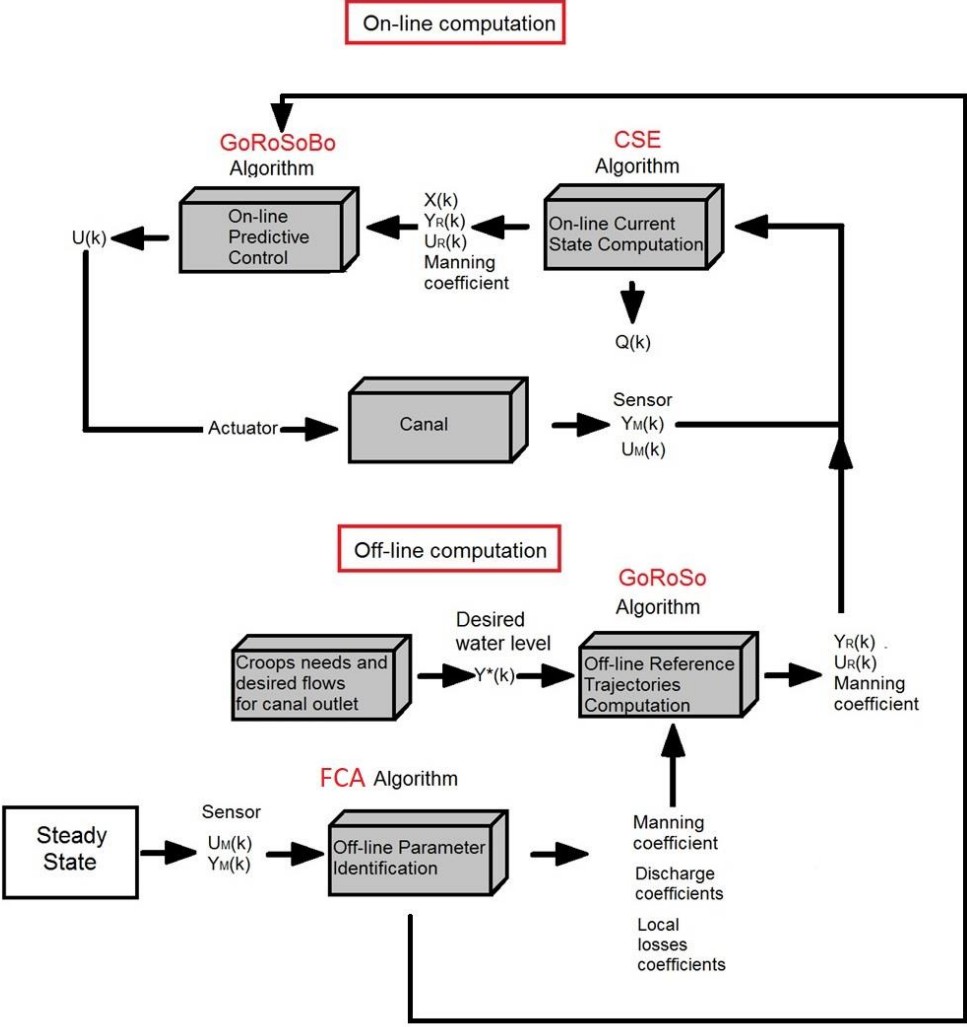

**Figure 2.** Overall control diagram of irrigation canal.

## 2. Materials and Methods

### 2.1. Control System Scheme

Control systems for irrigation canals were developed with the purpose of improving water operation tasks by providing the water demands at specific cross-sections in a period of time. In that sense, the main tools of a control system in irrigation canals are online and offline control algorithms, which are able to set operations (mainly gate movements) regarding water demands and redress pre-established operations in the case of unexpected disturbances (unknown flow disturbance or Manning roughness coefficient changes); see Figure 2.

All these operations are based on several tools and control algorithms, where all of them are synchronized, working together to fulfil a same objective, that is, water savings, efficiency, and sustainability. For this reason, the constitution of any modern online predictive control model requires the contribution of multiple different algorithms. The coordination and order of succession of the algorithms is indispensable for the performance of the off-line and online predictive control, since each algorithm develops a specific task, see [18,19].

The specific task and process developed by each algorithm in the overall control diagram is out of the scope of this paper (see E. Bonet Gil (2015) [19]).

In this paper, we focus on off-line parameter identification, because a small error in the estimation of any of these empirical parameters may introduce larger errors in the system.

Precisely, the objective of "Off-line Parameter Identification" algorithms is to avoid this kind of errors by regularly providing an updated estimation of their values.

«Crop needs and desired hydrographs for canal outlets»: the hydrographs at the lateral diversion points of the main canal are calculated on the basis of the water demands. They are fixed considering the farmer requirements and others demands accepted by the watermaster. The behaviour of the canal supplying these hydrographs determines the "desired behavior" (Y*) at several cross sections.

«Off-line Computation of the reference trajectories»: the desired behavior (Y*) must be transmitted to the "Reference Trajectories Calculation" algorithm that determines the positions of each gate. This algorithm calculates the optimum behavior (YR (reference water level)), which is the one most similar to the desired behavior that is physically possible. We call "UR" the optimum gate trajectories calculated to obtain the optimum behavior (YR). They must be calculated off-line (e.g., with an anticipated irrigation cycle). There is an extensive bibliography of feedforward control algorithms that compute the reference trajectories, such as GoRoSo [13].

The FC algorithm, presented in this paper, is conceived as an algorithm that forms part of the "Off-line Parameter Identification" process in the overall control diagram of an irrigation canal. In particular, out of all the existing physical parameters, the algorithm focuses on estimating the Manning roughness coefficient of irrigation canals, which has proven to be the most influential in terms of the sensitivity of the outcome of any control algorithm with respect to canal behavior.

The FC algorithm solves an inverse problem, shown in Equation (1), implemented as an unconstrained nonlinear optimization problem using the Levenberg–Marquardt method, whose solution is the partial derivatives of water level versus Manning roughness coefficients at several points in the canal, usually next to the canal offtakes.

$$
\begin{aligned}
\Delta Y &= [HIM'(n)]\Delta n \\
\Delta n &= [HIM'(n)]^{-1}\Delta Y \\
[HIM'(n)] &= \frac{\partial Y}{\partial n} \\
[HIM(n)] &= \left(\frac{\partial Y}{\partial n}, \frac{\partial V}{\partial n}\right)
\end{aligned}
\tag{1}
$$

where $\Delta Y$ represents the changes in water level at selected points of the canal, $\Delta n$ represents a change in the Manning's roughness coefficient, $HIM(n)$ is the hydraulic influence matrix that represents the influence of Manning's roughness coefficient on the water level and velocity along the canal, and $HIM'(n)$ is the simplified hydraulic influence matrix that represents the influence of a Manning's roughness coefficient on the water level at different points of the canal, see [18,19].

### 2.2. The HIM Matrix

The HIM matrix defines the influence of the Manning's roughness coefficient of any canal reach [20,21] over the hydraulic behaviour of canal cross-sections, generally limited to checkpoint sections. It is established using the full Saint-Venant equations, which are based on the conservation of mass and momentum and represent the governing equations in unsteady open canal flow. In their non-conservative form, they constitute a non-linear, second-order, hyperbolic system of partial differential equations (PDE), see [19].

As with any hyperbolic system, it can be transformed into its characteristic form. Such transformation of the Saint-Venant equations provides an ordinary system of four equations (2).

$$
\left.
\begin{aligned}
\frac{dv}{dt} + \frac{g}{c(y)}\frac{dy}{dt} &= g\left[S_0 - S_f(y,v)\right] \\
\frac{dv}{dt} - \frac{g}{c(y)}\frac{dy}{dt} &= g\left[S_0 - S_f(y,v)\right]
\end{aligned}
\right\}
\left.
\begin{aligned}
\frac{dx^+}{dt} &= v + c(y) \\
\frac{dx^-}{dt} &= v - c(y)
\end{aligned}
\right\}
\qquad
S_f(y,v) = n^2\frac{v|v|}{R_H^{\frac{4}{3}}} \quad c(y) = \sqrt{\frac{gA(y)}{T(y)}}
\tag{2}
$$

where $y$ is the level of the free surface with reference to the canal bottom, $v$ is the weighted average velocity of all the particles in a canal cross-section, $t$ is the time, $S_0$ is the canal bottom slope, $S_f(y, v)$ is the friction slope, and c is the celerity of a gravity wave, where $A(y)$ is the area of the wetted surface or a cross-section of the flow and $T(y)$ is the top width of the free surface.

Equation (2) cannot be solved analytically, thus, the use of numerical techniques becomes indispensable, with a wide range of methods being able to be used. In hopes of achieving the largest possible integration time-steps without loss of accuracy, a particular discretization in finite differences of the second order has been adopted, referred to as the discretization method of characteristic curves in [22]. Applying the cited method to Equation (2), and taking into account the characteristics curves that contain the points P–R and Q–R (Figure 3), the ensuing equations can be obtained.

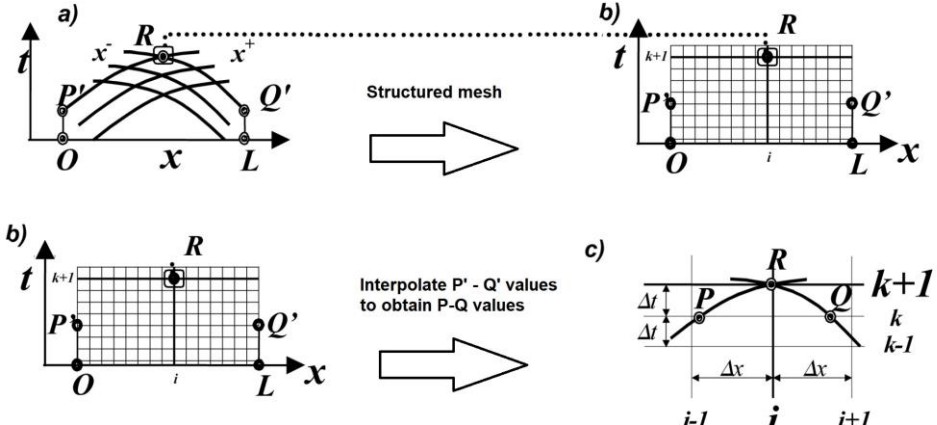

**Figure 3.** The steps for the interpolation onto a structured grid: (**a**) overlapping of the family of characteristic curves, (**b**) introducing the variables for every point (for instance P' and Q') to a structured mesh, (**c**) variables are interpolated at a particular mesh points P and Q.

The way of calculating the influences shown in this section is closely linked to the numerical scheme followed by the characteristic curves. However, this specific scheme is not employed, since it results in the solution at a point R whose coordinates $(x_R, t_R)$ are unknown a priori. These coordinates form part of the solution corresponding to the previous numerical scheme, although knowing the solution of the flow conditions at specific point and at specific time instants tends to be more useful, which, in the present case, would translate into treating $(x_R, t_R)$ as input variables as opposed to solution variables. In order to overcome this problem, a new approach centered on first interpolating and then solving the system is introduced (Figure 3).

A structured grid such as this one (Figure 3) creates a new nomenclature. Indeed, every variable is denoted by a double index, where $k$ refers to time and $i$ to space. As such, $y_{ik}$ and $v_{ik}$ represent the values for water level and average velocity at the coordinates $x_i = i\Delta x$ and $t_k = k\Delta t$ where $\Delta x$ and $\Delta t$ are selected by the user.

Furthermore, there are many control structures in canals. The individual study of each one is not feasible in this work, hence, only the most usual structures are introduced. A common characteristic is a checkpoint structure (Figure 4), which is a target point where the water level is measured with a depth gage, and it comprises a sluicegate, a lateral weir outlet, and an offtake orifice or a pump (shown in [23,24]). The interaction of this control

structure with the flow may be described according to the mass and energy conservation equations, shown in Equation (3).

$$\begin{aligned}
S(y_e)\frac{dy_e}{dt} &= A(y_e)v_e - q_b - q_s(y_e) - A(y_s)v_s - q_{offtake}(y_e) \\
A(y_s)v_s &= k_c u \sqrt{y_e - y_s + d} \\
q_s(y_e) &= C_S a_S (y_e - y_0)^{\frac{3}{2}} \\
q_{offtake}(y_e) &= C_0 A_0 \sqrt{2g y_e} \\
k_c &= \sqrt{2g} C_d a_c
\end{aligned} \tag{3}$$

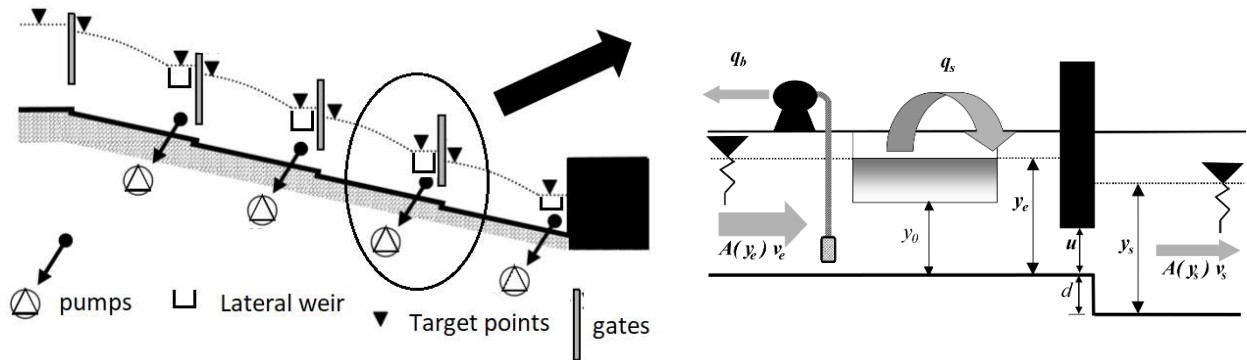

**Figure 4.** Example of a canal profile and diagram of a checkpoint with gate, lateral weir, and pump.

- $v_e$ is the weighted average velocity of all the particles in a canal cross-section;
- $y_e$ is the water level of all the particles in a canal cross-section;
- $S(y_e)$ is the horizontal surface of the reception area in the checkpoint;
- $A(y_e)*v_e$ is the incoming flow to checkpoint, defined in terms of water level and velocity;
- $A(y_s)*v_s$ is the outflow from the checkpoint that continues along the canal, described in terms of water level and velocity;
- $C_d$ is the discharge coefficient of the sluicegate and ac is the sluicegate width;
- $d$ is the checkpoint drop, and u is the gate opening;
- $q_b$ is the pumping offtake;
- $q_s(ye)$ is the outgoing lateral flow through the weir where $Cs$ is the discharge coefficient, as is the weir width, and $y_0$ is the weir height measured from the bottom, called weir equation;
- $Q_{offtake}(y_e)$ is the outflow orifice flow where $C_0$ is the discharge coefficient, $A_0$ is the area of the offtake orifice, called orifice offtake equation;
- $u$ is the open gate height

The presence of checkpoints (target points) or control structures along the canal leads to its sub-division into canal pools, in a way that there is always a canal pool between two checkpoints, and there is a checkpoint between two pools. By discretizing the control structure equations in a structured grid, taking into account the previous characteristic Equation (2) and adopting a common nomenclature, the control structure Equation (3) are re-written as indicated in the following system, composed of six equations, shown in Equation (4); see Figure 5.

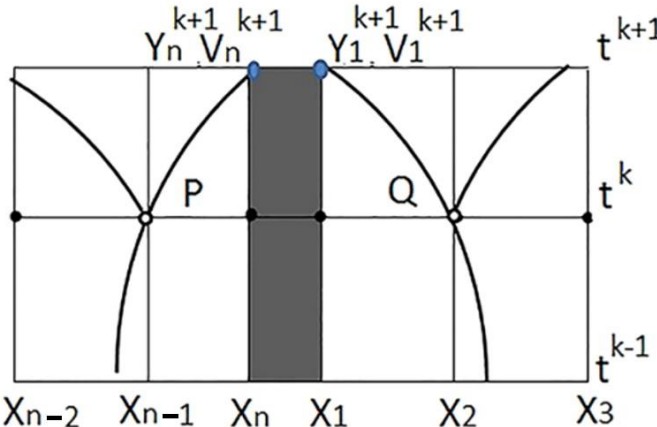

**Figure 5.** Graph with discretization of the control structure equation.

In that way, $y_n^{k+1}$ represents the water level at node n at the section upstream of the control structure at time $k + 1$, that is, the incoming water level $y_i$. In the same way, $y_1^{k+1}$ is defined as the existing water level at the first node of the downstream pool from the checkpoint at the same time $k + 1$, and $y_o$ the outgoing water level at the control structure. The same terminology reasoning may be applied for the velocities $v_n^{k+1}$ and $v_1^{k+1}$.

$$\left.\begin{aligned}
&f_1 \equiv x_n - x_P - \tfrac{1}{2}\Delta t \left[ v_n^{k+1} + c_n^{k+1} + v_P + c_P \right] = 0 \\
&f_2 \equiv \left( v_n^{k+1} - v_P \right) + \tfrac{g}{2} \tfrac{c_n^{k+1} + c_P}{c_n^{k+1} c_P} \left( y_n^{k+1} - y_P \right) - g\Delta t \left( \tfrac{S_{f_n}^{k+1} + S_{f_P}}{2} - S_0 \right) = 0 \\
&f_3 \equiv \left( v_1^{k+1} - v_Q \right) - \tfrac{g}{2} \tfrac{c_1^{k+1} + c_Q}{c_1^{k+1} c_Q} \left( y_1^{k+1} - y_Q \right) - g\Delta t \left( \tfrac{S_{f_1}^{k+1} + S_{f_Q}}{2} - S_0 \right) = 0 \\
&f_4 \equiv x_1^{k+1} - x_Q - \tfrac{1}{2}\Delta t \left[ v_1^{k+1} - c_1^{k+1} + v_Q - c_Q \right] = 0 \\
&f_5 \equiv A\left( y_n^{k+1} \right) v_n^{k+1} - q_b - q_s\left( y_n^{k+1} \right) - A\left( y_1^{k+1} \right) v_1^{k+1} - q_{offtake}\left( y_n^{k+1} \right) = 0 \\
&f_6 \equiv A\left( y_1^{k+1} \right) v_1^{k+1} - k_c u \sqrt{y_n^{k+1} - y_1^{k+1} + d} = 0
\end{aligned}\right\} \quad (4)$$

where
$\Delta t = t^{k+1} - t^P = t^{k+1} - t^Q;$
$y_P(x_P) = s(x_P, y_{n-2}^k, y_{n-1}^k, y_n^k); \; y_Q(x_Q) = s(x_Q, y_1^k, y_2^k, y_3^k);$
$v_P(x_P) = s(v_P, v_{n-2}^k, v_{n-1}^k, v_n^k); \; v_Q(x_Q) = s(x_Q, v_1^k, v_2^k, v_3^k);$
$c_n^{k+1} = c(y_n^{k+1}); \; c_1^{k+1} = c(y_1^{k+1});$
$S_{f_n}^{k+1} = S_f(y_n^{k+1}, v_n^{k+1}); \; S_{f_1}^{k+1} = S_f(y_1^{k+1}, v_1^{k+1}).$

On the other hand, $x_P$, $y_n^{k+1}$, $v_n^{k+1}$, $y_1^{k+1}$, $v_1^{k+1}$, and $x_Q$ remain as the unknown variables of the problem, which consists of finding the influences of the Manning roughness coefficient (n) on the flow conditions along the canal.

In (5), the Manning roughness coefficient (n) explicitly appears in the description for the first time. Despite the fact that the specific form of this function is still unknown, (5) shows that the influence of the parameter *n* on flow conditions at time $k + 1$ is the sum of the indirect influence of the conditions at instant k and the direct influence at instant $k + 1$ through the term "*L*", which represents the variation in the Manning roughness coefficient.

Here

$$M\frac{\partial}{\partial n}\begin{pmatrix} x_P \\ y_1^{k+1} \\ v_1^{k+1} \\ y_n^{k+1} \\ v_n^{k+1} \\ x_Q \end{pmatrix} = NS\frac{\partial}{\partial n}\begin{pmatrix} y_{n-2}^k \\ v_{n-2}^k \\ y_{n-1}^k \\ v_{n-1}^k \\ y_n^k \\ v_n^k \\ y_1^k \\ v_1^k \\ y_2^k \\ v_2^k \\ y_3^k \\ v_3^k \end{pmatrix} + L \tag{5}$$

where

$M = \dfrac{\partial(f_1,f_2,f_3,f_4,f_5,f_6)}{\partial\left(x_P,y_n^{k+1},v_n^{k+1},y_1^{k+1},v_1^{k+1},x_Q\right)}$

$N = -\dfrac{\partial(f_1,f_2,f_3,f_4,f_5,f_6)}{\partial\left(x_P,y_P,v_P,y_Q,v_Q,x_Q\right)}$

$L = \begin{pmatrix} 0 & \frac{\partial f_2}{\partial n} & \frac{\partial f_3}{\partial n} & 0 & 0 & 0 \end{pmatrix}^T$

$S = \dfrac{\partial\left(x_P,y_P,v_P,y_Q,v_Q,x_Q\right)}{\partial\left(x_P,y_{n-2}^k,v_{n-2}^k,y_{n-1}^k,v_{n-1}^k,y_n^k,v_n^k,y_1^k,v_1^k,y_2^k,v_2^k,y_3^k,v_3^k,x_Q\right)}$

As a summary, the method of characteristics is applied to the Saint-Venant equations in order to obtain a set of algebraic equations that establish a relationship between the influence parameter n and the hydrodynamic canal state, lumping all the influences together in a global matrix, which is referred to as *HIM*(*n*). Based on this system of equations and employing the first derivative (∂y/∂n, ∂v/∂n) in an analytical process, the changes in flow behavior (water level and velocity) due to a change in Manning's roughness coefficient at a point at a certain time instant may be determined.

### 2.3. The Optimization Problem

The inverse problem shown in Equation (1) is formulated as an unconstrained optimization problem. It is the classical non-linear least-squares problem without constraints and is solved, in this case, by means of the Levenberg–Marquardt method, which consists of an iterative algorithm.

To introduce the optimization problem, some vectors used in the development must be evaluated. As explained before, the FC algorithm requires, as input data, the water level measured at some points (checkpoints) for a past-time horizon fixed by the watermaster. Now, consider a vector (measured water-level vector), which contains the water-level measurements at the checkpoints from the time instant 1 to $k_F$ (6) whose dimension is $n_y$, where $n_y = k_F \times n_c$, where $k_F$ is the final instant of the past-time horizon, and $n_c$ is the number of checkpoints. The measured water-level vector is then defined as:

$$Y^* = \left[y_1^*(1), y_1^*(2), \ldots, y_{n_c}^*(k_F - 1), y_{n_c}^*(k_F)\right]^T \tag{6}$$

The corresponding values to this vector may be checked in a computational grid in Figure 6 (big red dots).

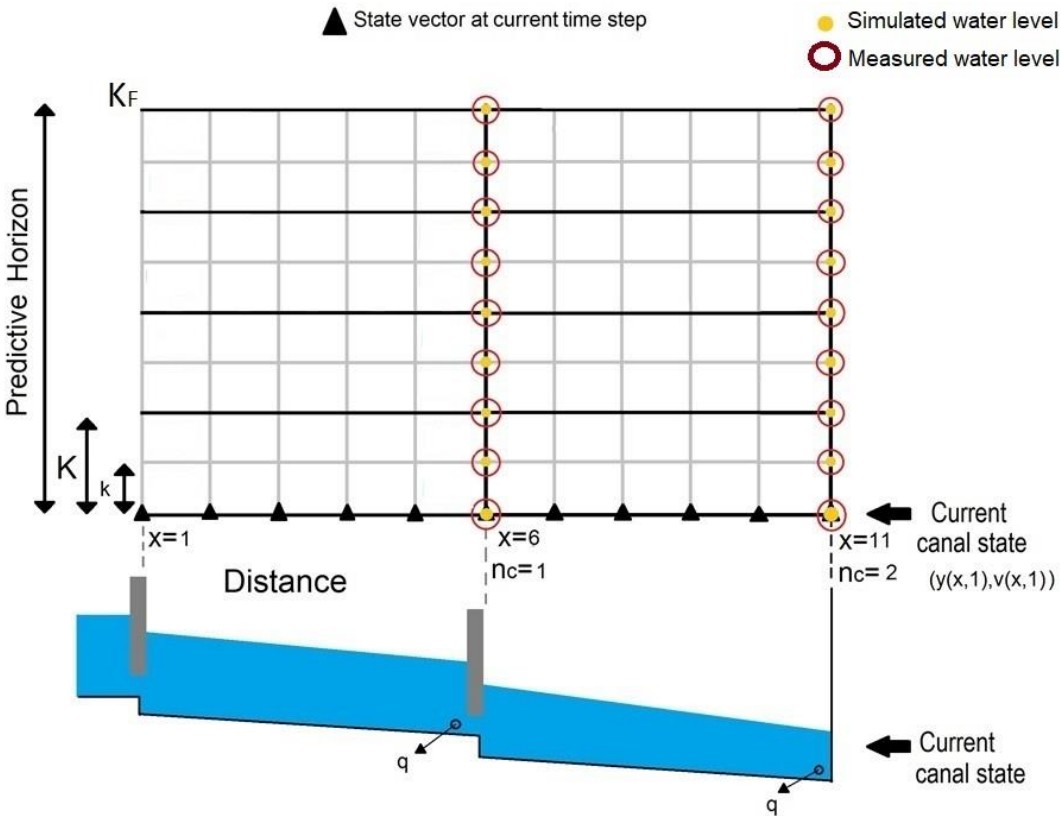

**Figure 6.** Sketch of a numerical grid of a canal with two pools controlled by two checkpoints downstream each pool. Also, it shows the x/t-dots where the flow behaviour is defined. Notice that "K" with capital letter denotes time interval of control and "k" with small letter denotes time instant of simulation.

Similarly, the "predicted output vector" $Y(k)$ may be obtained, which is defined as the vector containing the simulating water level (small yellow dots) from the output data predicted by the FC algorithm at the time instant k for all the discretization points in the canal:

$$Y_1^{k_F} = [y_1(1), y_1(2), \ldots, y_{n_c}(k_F - 1), y_{n_c}(k_F)]^T \tag{7}$$

The predicted output vector at the current time defines the simulated water level in a computational grid in Figure 6 (small yellow points).

Due to the fact the optimization process only considers the differences in water level between the measured and simulated values at target points, it is necessary to gather these corresponding values resulting from the algorithm.

As previously anticipated, the FC algorithm calculates the resulting Manning roughness coefficient trajectories at several points (for instance, canal pools) during a past-time horizon. In that case, as illustrated in Figure 7, it is assumed that the friction coefficient of a canal pool value may be susceptible to variations at every operation period K. In that way, the Manning roughness coefficient trajectories can be approached by piecewise functions. The Manning roughness coefficient trajectories vector is defined by lumping together all the roughness coefficient trajectories during the past-time horizon, as follows:

$$n = \left[ n_1(1), \ldots, n_{n_p}(1), \ldots, \ldots, n_1(K_F), \ldots, n_{n_p}(K_F) \right]^T \tag{8}$$

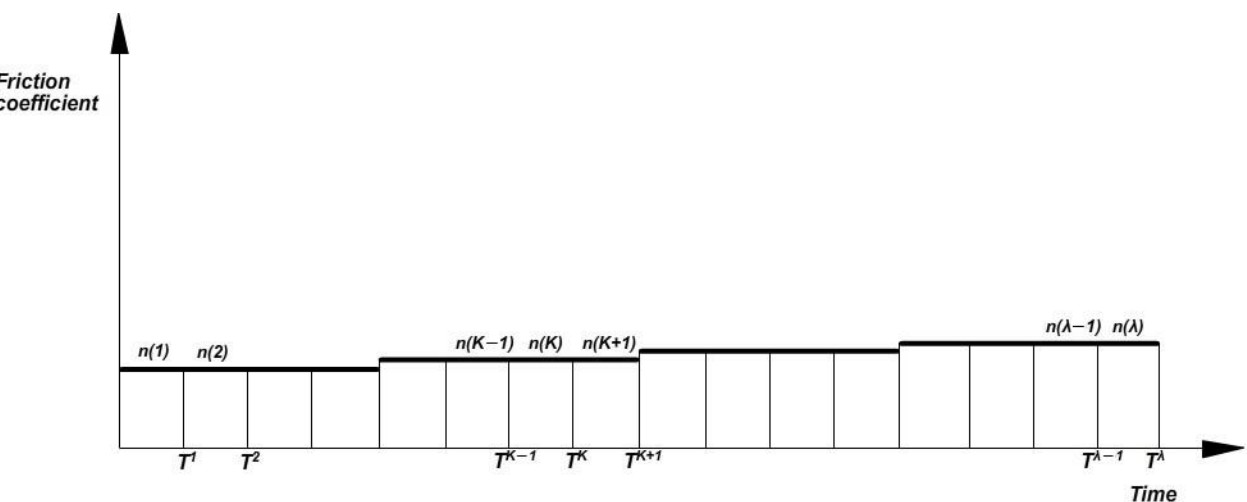

**Figure 7.** Mathematical representation of a Manning roughness coefficient trajectory.

With the dimension of this vector being $n_n = n_p \times K_F$, where $n_p$ is the number of canal pools and $K_F$ is the final operation period of the past-time horizon.

To sum up, the FC algorithm calculates the Manning roughness trajectories ($\Delta n$), which reduce the error ($\Delta Y$) between measured and simulated water levels at specific cross-section in a period of time.

In terms of the optimization problem, the objective is to make the simulated water-level vector as similar as possible to the measured water-level vector by manipulating the Manning roughness coefficient trajectories vector, see [24,25]. In mathematical terms, the objective is to obtain the Manning roughness coefficient trajectories vector ($n$) that minimizes the following performance criterion:

$$Minimize\ J(n) = \frac{1}{2}(Y_1^{K_F}(n) - Y^*)^T [Q]\left(Y_1^{K_F}(n) - Y^*\right) \tag{9}$$

where $J(n)$ is the objective function, $Y(n)$ is the prediction output vector, $Y^*$ is the measured water-level vector, $Q$ is a weighting matrix, and $n$ contains the Manning roughness coefficient trajectories (8).

### 3. Practical Examples, Results, and Discussion

*3.1. Practical Example: A Canal with Two Pools*

In this example, we proposed several scenarios in order to test the FC algorithm in a canal that has two pools separated by sluicegates (Figure 8). The flow is controlled by a gate downstream from the reservoir. Water is delivered through gravity outlets at the downstream end of each pool, where the checkpoints are located. There are pumping stations at the end of each pool, which can introduce disturbances in the system in space and time.

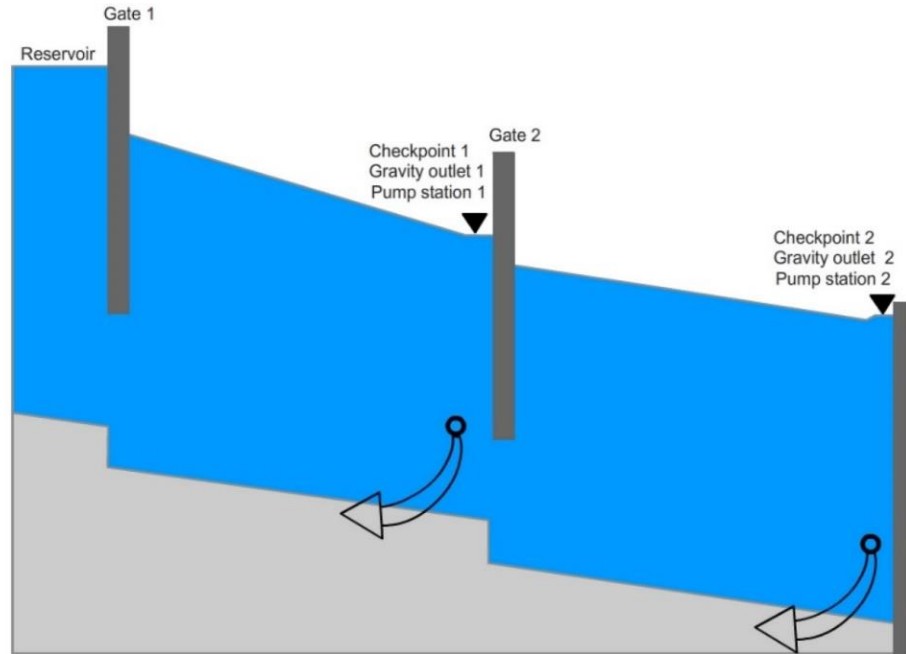

**Figure 8.** Canal profile of a canal with two pools (the arrows represent the outlet flow for the offtake orifice).

The canal with a trapezoidal section is represented in Figure 8, and the general data (Mannings coefficient, canal depth, pool length, bottom slope, side slopes, and bottom width) are shown in Table 1. The characteristics of checkpoint, sluicegate, pump station, and orifice offtake are shown in Table 2:

**Table 1.** General canal features.

| Pool Number | Pool Length (km) | Bottom Slope (%) | Side Slopes (H:V) | Manning's Coefficient (n) | Bottom Width (m) | Canal Depth (m) |
|---|---|---|---|---|---|---|
| I | 2.5 | 0.1 | 1.5:1 | 0.025 | 1 | 2.5 |
| II | 2.5 | 0.1 | 1.5:1 | 0.025 | 1 | 2.5 |

**Table 2.** Checkpoints and sluicegate/pump station/orifice offtake features (control structure).

| Number of Control Structure or Checkpoint | Gate Discharge Coefficient | Gate Width (m) | Gate Height (m) | Step (m) | Discharge Coef./Diameter Orifice Offtake (m) | Orifice Offtake Height (m) | Lateral Spillway Height (m) | Lateral Spillway Width (m)/Discharge Coefficient |
|---|---|---|---|---|---|---|---|---|
| 0 | 0.61 | 5.0 | 2.5 | 0.6 | - | - | - | - |
| 1 | 0.61 | 5.0 | 2.5 | 0.6 | 0.6/0.77 | 1.0 | 2.3 | 500/1.99 |
| 2 | - | - | - | - | 0.6/0.77 | 1.0 | 2.3 | 500/1.99 |

In this test, an upstream large reservoir is considered, whose water level $H_{reservoir}$ is constantly 3 m throughout the test. At the end of the last pool, there is a control structure with an orifice offtake and a pump station. The flow through the orifice offtake depends on the upstream water level of the orifice and the gate position and water demands depend on the scenario. In any case, there is an orifice offtake, pump station, and checkpoint at the end of every pool. This example starts from an initial steady state, where the upstream boundary condition is a flow rate through gate one of 10 m³/s, and the checkpoint condition (at the end of each pool) is a water level of 2.0 m, which involves a constant flow rate (scheduled demand) of 5 m³/s by the orifice offtake regarding the initial canal conditions (Manning's

coefficient [26,27], gate position, etc.). Furthermore, the water level is measured at every checkpoint every 5 min during an operational horizon of 4 h.

These initial conditions are established in a way that an initial steady state is exhibited in the canal considering a Manning's roughness coefficient equal to $n_{obj} = 0.025$ for both pools.

The main objective on this test is to check the FC algorithm performances and accuracy. For this reason, we test a two-pool canal using the FC algorithm in order to identify its real Manning coefficient ($n_{obj} = 0.025$). This test is performed using two different scenarios in an irrigation canal under unsteady flow conditions, as irrigation canals are frequently operating under unsteady flow conditions so the algorithm must be able to obtain the Manning coefficient under these conditions. In that sense, we are testing two scenarios. In the first, the gate position changes during the operating time horizon and in the second, that water demand changes during the operating time horizon.

In order to evaluate the accuracy of the FC algorithm, we use a performance indicator that examines the deviations in the absolute value between those calculated by the algorithm and the objective Manning's roughness coefficient trajectories ($n$ and $n_{obj}$, respectively), which is mathematically expressed as the norm of the vector difference:

$$\varepsilon(n) = \left\| n - n_{obj} \right\| \tag{10}$$

### 3.1.1. Scenario One: Changes in Gate Position

This scenario stars from an initial canal steady state. However, this steady state is disrupted by the introduction of a known disturbance in the canal by changing the initial gate position (gate one) during the simulating time horizon of 4 h. In particular, the opening of the first gate of the canal is increased by 30% for 25 min, from minute 45 until minute 70, while the opening of the second gate remains in the same position during the simulating time horizon.

The initial Manning's roughness coefficient trajectories considered by the FC algorithm is 0.035 during the simulating time horizon.

### Results

The water level measurements at checkpoints resulting from scenario one is shown in Figure 9. Up until minute 45, the canal presents the initial steady state. Then, the sluicegate located at the upstream node of the canal opens up more, which increases the water level from 2 m to 2.21 m at checkpoint one.

### 3.1.2. Scenario One: Results

After completing the input of the inverse problem (water level measurements, gate trajectories, pump flow trajectories, and scheduled deliveries), the FC algorithm is tested regarding scenario one. The results of the Manning's roughness coefficient trajectories predicted by the algorithm are displayed in Figure 10.

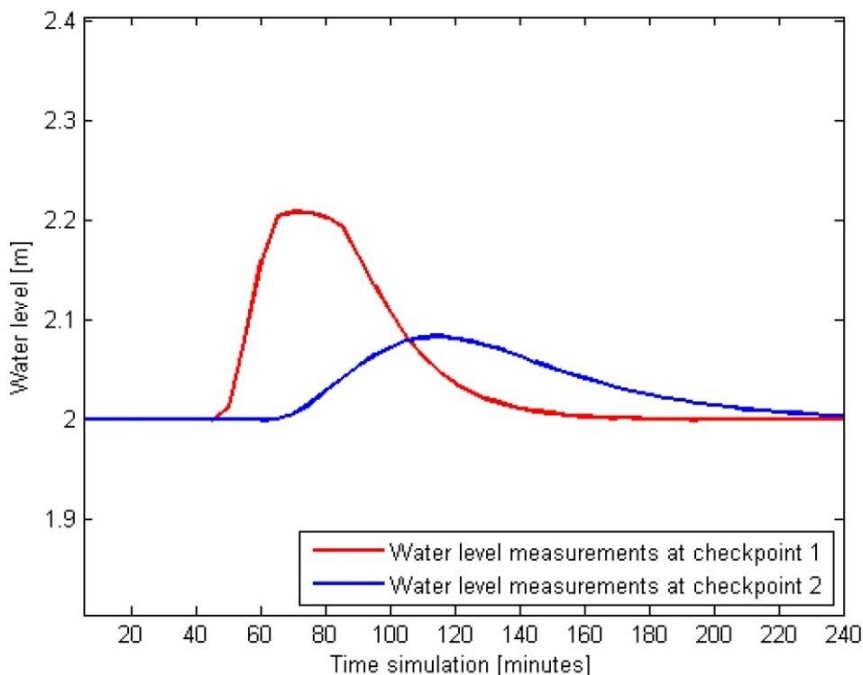

**Figure 9.** Water level at the checkpoints one (red line) and two (blue line) in scenario one.

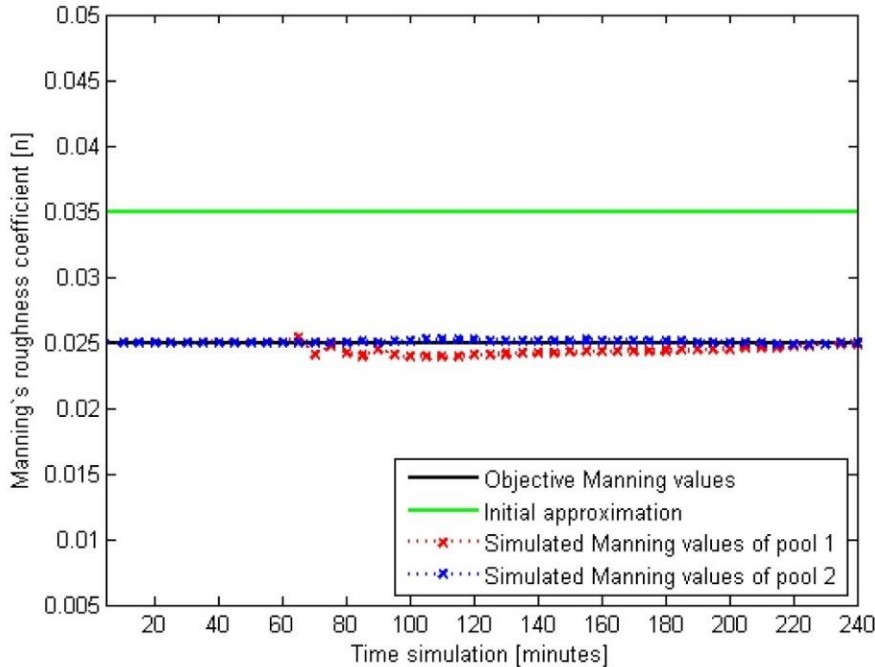

**Figure 10.** Manning's roughness coefficient trajectories at pools one (red line) and two (blue line) in scenario one.

The estimated Manning's roughness coefficient trajectory corresponding to the second pool fits exactly with the real trajectory for all regulation periods. Instead, the determined Manning's roughness coefficient trajectory of the first pool does exhibit slight differences with respect to the real values of $n_{obj}$ = 0.025 during the middle part of the simulation, although the biggest Manning coefficient deviation (among real and calculated) is around $1 \times 10^{-3}$ and the average Manning roughness coefficient for the second canal pool is 0.02498 (that is, a Manning coefficient deviation of $2 \times 10^{-4}$). In this scenario, the performance indicator value ($\varepsilon(n)$) is equal to $4.69 \times 10^{-3}$.

### 3.1.3. Scenario Two: Changes in Water Demand

The following scenario is built upon the premise that the scheduled deliveries of water are no longer constant throughout the simulation while the rest of variables remain the same. The water demand through the first gravity outlet increases from 5 m$^3$/s to 6.5 m$^3$/s, only from minute 45 until minute 70, which is also the time interval used in the previous scenario.

The initial Manning's roughness coefficient trajectory considered by the FC algorithm is 0.035 during the simulating time horizon.

The water level at the first checkpoint is 2 m until the scheduled delivery suddenly increases at minute 45 due to changes in water demand. From there, the water level drops down to a minimum of 1.73 m and quickly starts to recover once the water demand reverts to 5 m$^3$/s, returning to the initial steady state (see Figure 11).

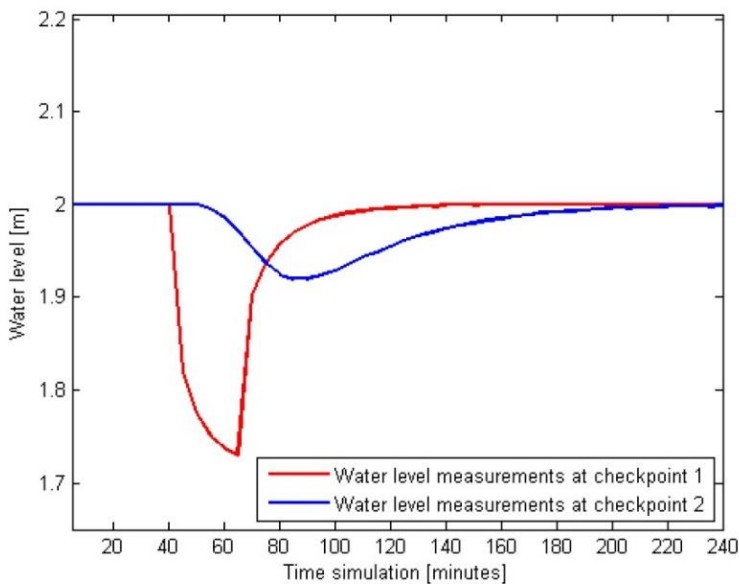

**Figure 11.** Water level at the checkpoints one (red line) and two (blue line) in scenario two.

### 3.1.4. Scenario Two: Results

After completing the input of the inverse problem (water level measurements, gate trajectories, pump flow trajectories, and scheduled deliveries), the FC algorithm is tested in scenario two. The results of the Manning's roughness coefficient trajectories predicted by the algorithm are displayed in Figure 12.

The FC algorithm is nearly able to produce perfect results as both pool trajectories show values very close to $n_{obj} = 0.025$. The biggest Manning coefficient deviation is lower than $1 \times 10^{-3}$ and the average Manning coefficient for the second canal pool is almost the objective Manning's roughness coefficient. In scenario two, the performance indicator ($\varepsilon(n)$) of the Manning's roughness coefficient is equal to $2.691 \times 10^{-3}$. Therefore, the FC algorithm is able to obtain the Manning's roughness coefficient for the two-pool canal with a high accuracy.

The robustness of the algorithm is quite high because the results in every different scenario show accurate Manning's roughness coefficients. In any case, the average Manning's roughness coefficient deviations (among real and calculated) only introduce water level errors at the cross-sections that are less than 2 mm.

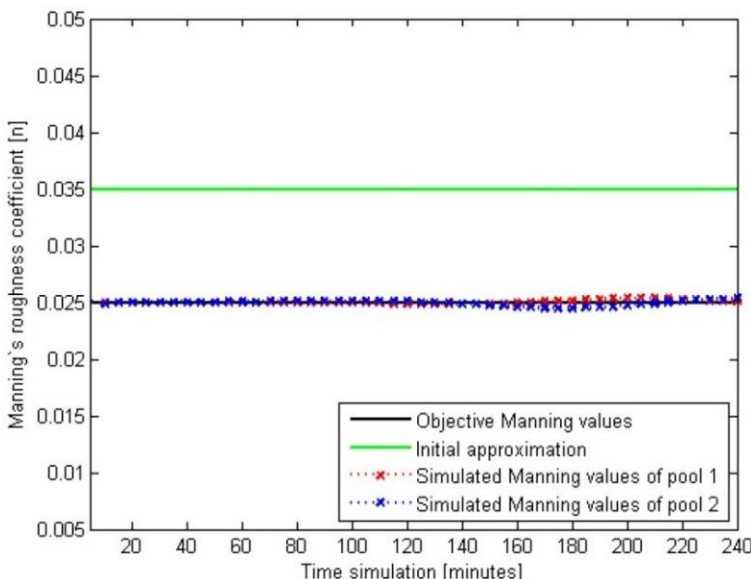

**Figure 12.** Manning's roughness coefficient trajectories at pools one (red line) and two (blue line) in scenario two.

### 3.2. Practical Example: A Canal with Multiple Disturbances at the Same Time: ASCE TEST CASE

In this practical example, we introduce the test cases that were proposed by the ASCE committee to evaluate control algorithms [19]. These test cases were originally devised to evaluate feedforward and feedback control algorithms that recalculate the gate trajectories at each regulation period in a predictive horizon in order to adjust the water levels at the checkpoints. In spite of the fact that this is not the aim of the FC algorithm, the FC algorithm should be able to obtain the most accurate Manning's roughness coefficient, as these results will support a feedback or feedforward control algorithm, such as GoRoSo [13] and GoRoSoBo simplified [27], which would calculate the most accurate gate trajectories from the most accurate Manning's roughness coefficient.

In that sense, the FC algorithm estimates Manning's n values based on the known basic canal control variables for a past-time horizon: scheduled deliveries, gate trajectories, pump flow trajectories, and water-level measurements at the checkpoints of the canal.

In particular, the performance of the FC algorithm is assessed in one of the previous test cases suggested by the ASCE. The selected test case, denoted as test 2-1 for Example Canal 2 in [17], considers the introduction of a series of scheduled and unscheduled flow disturbances in the Corning canal, which is one of the two canals contemplated by the ASCE. The gate trajectory parameters, which are now assumed to be an input parameter instead of an output parameter, will be obtained from the available results of feedback real-time controllers [27] that have already performed the original test case 2-1 for the Corning canal.

#### 3.2.1. ASCE TEST CASE: Canal Features

The Corning canal tested herein is based on the upstream portion of the real Corning Canal in California, which is characterized by being long, having a mild slope, and presenting a significant storage capacity. The length of the canal considered is 28 km and its cross-sections are trapezoidal. It is divided into eight pools that are separated by eight rectangular gates and delimited by a total of nine points (identified by numbers from 0 to 8). The first point (0) is not a checkpoint as it solely contains the first sluicegate that separates the canal from a large reservoir. Other hydraulic structures along the canal include orifice offtakes, emergency lateral spillways, and pump stations, which are all found at

the checkpoints or targets points located at the downstream end of each pool where the water-level measurements are taken.

The canal geometry is shown in Figure 13 along with the number scheme of the pools and nodes employed, while the general features of the canal pools are represented in Table 3 and the characteristics of the hydraulic control structures at each node are displayed in Table 4.

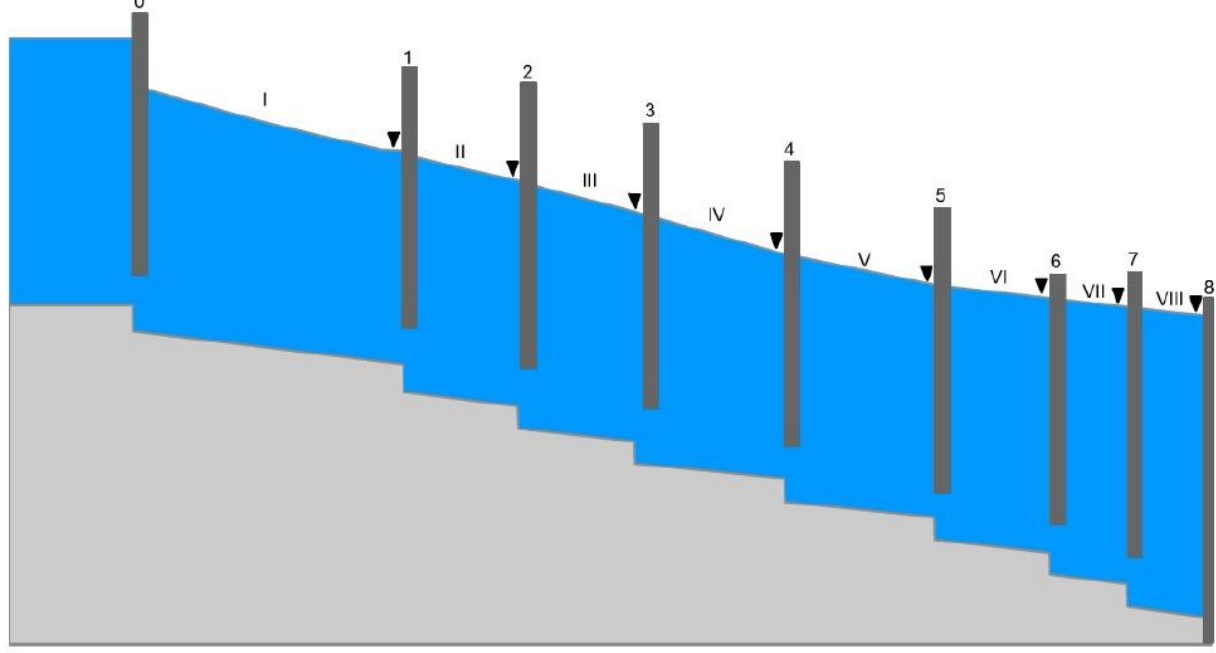

**Figure 13.** Corning canal profile. The red lines mark the position of checkpoints. The first pool is number I (so every pool in the canal has been enumerated (I–VIII)), the first checkpoint is number 1 (so every checkpoint in the canal has been enumerated (1–8)) and the triangles represent the target water level.

**Table 3.** Features of Corning canal pools.

| Pool Number | Pool Length (Km) | Bottom Slope | Side Slopes (H:V) | Manning's Coefficient (n) | Bottom Width (m) | Canal Depth (m) |
|---|---|---|---|---|---|---|
| I | 7 | $10 \times 10^{-4}$ | 1.5:1 | 0.02 | 7 | 2.5 |
| II | 3 | $10 \times 10^{-4}$ | 1.5:1 | 0.02 | 7 | 2.5 |
| III | 3 | $10 \times 10^{-4}$ | 1.5:1 | 0.02 | 7 | 2.5 |
| IV | 4 | $10 \times 10^{-4}$ | 1.5:1 | 0.02 | 6 | 2.3 |
| V | 4 | $10 \times 10^{-4}$ | 1.5:1 | 0.02 | 6 | 2.3 |
| VI | 3 | $10 \times 10^{-4}$ | 1.5:1 | 0.02 | 5 | 2.3 |
| VII | 2 | $10 \times 10^{-4}$ | 1.5:1 | 0.02 | 5 | 1.9 |
| VIII | 2 | $10 \times 10^{-4}$ | 1.5:1 | 0.02 | 5 | 1.9 |

As shown in Table 3, the Manning's roughness coefficient value of each pool in the Corning canal is equal to n = 0.020. Therefore, the objective Manning's n value that must be achieved by the FC algorithm is $n_{obj}$ = 0.020 for the eight pools considered. At the start of the simulation run, Manning's n value will be assumed to be equal to an initial guess value, equal to 0.035.

**Table 4.** Corning canal control structures.

| Target Points | Gate Discharge Coefficient | Gate Width (m) | Gate Height (m) | Step (m) | Length from Gate 1 (km) | Orifice Offtake Height (m) | Lateral Spillway Height (m) |
|---|---|---|---|---|---|---|---|
| 0 | 0.61 | 7 | 2.3 | 0.2 | 0 | - | 3 |
| 1 | 0.61 | 7 | 2.3 | 0.2 | 7 | 1.05 | 2.5 |
| 2 | 0.61 | 7 | 2.3 | 0.2 | 10 | 1.05 | 2.5 |
| 3 | 0.61 | 7 | 2.3 | 0.2 | 13 | 1.05 | 2.5 |
| 4 | 0.61 | 6 | 2.1 | 0.2 | 17 | 0.95 | 2.3 |
| 5 | 0.61 | 6 | 2.1 | 0.2 | 21 | 0.95 | 2.3 |
| 6 | 0.61 | 5 | 1.8 | 0.2 | 24 | 0.85 | 1.9 |
| 7 | 0.61 | 5 | 1.8 | 0.2 | 26 | 0.85 | 1.9 |
| 8 | - | - | - | - | 28 | 0.85 | 1.9 |

The past-time horizon of the simulation is 12 h (43,200 s), as prescribed by the ASCE for test cases 2-1 regarding the second part of a complete irrigation cycle (24 h). Once the overall control diagram (on-line computation) has completed an irrigation cycle (so we know the gate trajectories, scheduled and unscheduled water demands, etc.) regarding output values from GoRoSoBo simplified and CSE, it is time to check the Manning's roughness coefficient of the different pools of the canal during the last irrigation cycle. In that sense, the FC algorithm calculates the Manning's roughness coefficient trajectories for the second part (12 h) of a complete irrigation cycle. In this manner, the Manning's roughness coefficient trajectories will be checked for the next irrigation cycle and introduced in off-line and on-line controllers. The time step between successive control actions is determined to be equal to T = 900 s (15 min).

### 3.2.2. ASCE TEST CASE: Initial and Boundary Conditions

The upper boundary condition is established by the constant water level provided by the large reservoir upstream of the first gate in the Corning canal, which is imposed to be equal to 3 m over the entire simulation. On the other hand, the boundary conditions for every pool are given by the discharge through the orifice offtake and the flow extracted by the pump, both of which are located downstream of the eighth pool.

The pump at the end of the canal extracts a constant water flow of 3 m$^3$/s. The initial conditions of the test case, shown in Table 5, correspond to the initial steady state of the canal.

**Table 5.** Initial and unscheduled offtake changes in test case 2-1.

| Pool Number | Offtake Initial Flow (m$^3$/s) | Check Initial Flow (m$^3$/s) | Unscheduled Offtake Changes at 2 h (m$^3$/s) | Check Final Flow (m$^3$/s) |
|---|---|---|---|---|
| Heading | - | 13.5 | - | 11.5 |
| I | 1.0 | 12.5 | - | 10.5 |
| II | 1.0 | 11.5 | - | 9.5 |
| III | 1.0 | 10.5 | - | 8.5 |
| IV | 1.0 | 9.5 | - | 7.5 |
| V | 2.5 | 7.0 | - | 5.0 |
| VI | 2.0 | 5.0 | −2.0 | 5.0 |
| VII | 1.0 | 4.0 | - | 4.0 |
| VIII | 1.0 | 3.0 | - | 3.0 |

With respect to the offtake flows, the initial and unscheduled changes considered in test case 2-1 are also presented in Table 5.

Furthermore, the canal is required to maintain certain values of water levels at the checkpoints so that the flows through the orifice offtakes match exactly the scheduled delivery demands during the whole irrigation cycle, which in this test case that lasts 12 h.

The desired water levels to be accomplished in the canal are revealed in Table 6 for each checkpoint in the Corning canal, and they constitute the water-level measurements of the analyzed past-time horizon.

**Table 6.** Target depth values in test case 2-1.

| Checkpoint | Target Water Level (m) |
| --- | --- |
| Heading | - |
| I | 2.1 |
| II | 2.1 |
| III | 2.1 |
| IV | 1.9 |
| V | 1.9 |
| VI | 1.7 |
| VII | 1.7 |
| VIII | 1.7 |

### 3.2.3. ASCE TEST CASE: Scenario

The sluicegate trajectories were obtained from GoRoSoSo simplified [27], that is, an online controller across the predictive horizon, and the unscheduled water demands were obtained from CSE algorithm [9] regarding the overall control diagram.

As anticipated before, many feedback control algorithms have undergone the simulation of test 2-1 for the Corning canal, highlighting amongst them the performance of GoRoSoBo simplified, which, in cooperation with CSE, was able to produce very competent gate trajectory results for this test case [19].

The gate trajectories obtained by GoRoSoBo simplified in test 2-1 for the Corning canal are shown in Figures 14 and 15, from which the complexity and extent of the gate movements required in each pool to maintain the target values at the checkpoints at all times due to the introduction of unscheduled off-take changes can be appreciated. These gate trajectories are, by a large margin, the most complicated gate movements in respect to last example, both in terms of number of parameters, number of pools, pumps, etc.

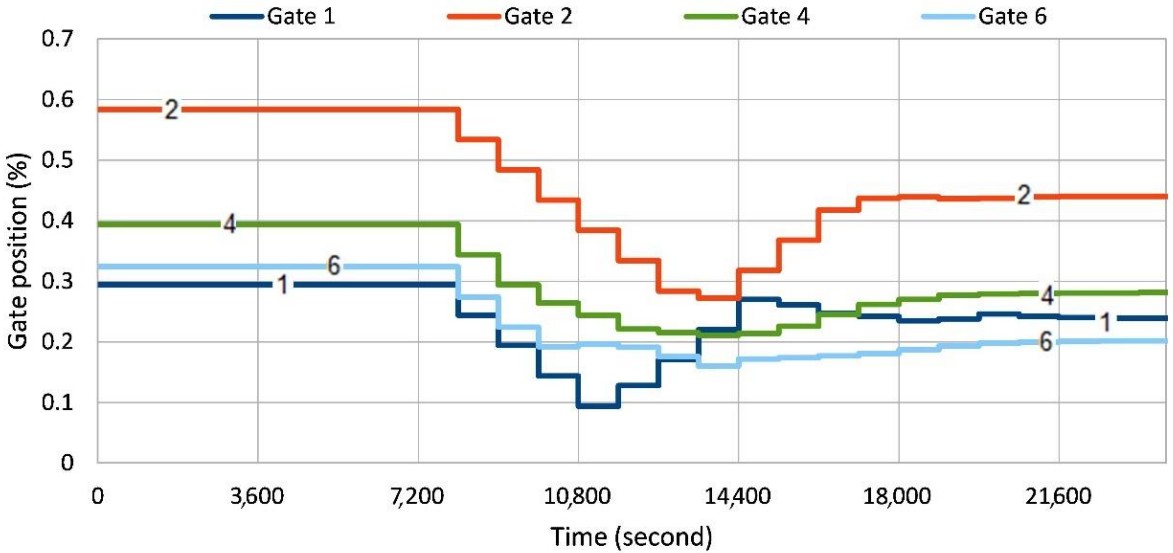

**Figure 14.** Sluice-gate trajectories obtained by GoRoSoBo Simplified in Test 2-1 for the Corning canal (Gates 1-2-4-6).

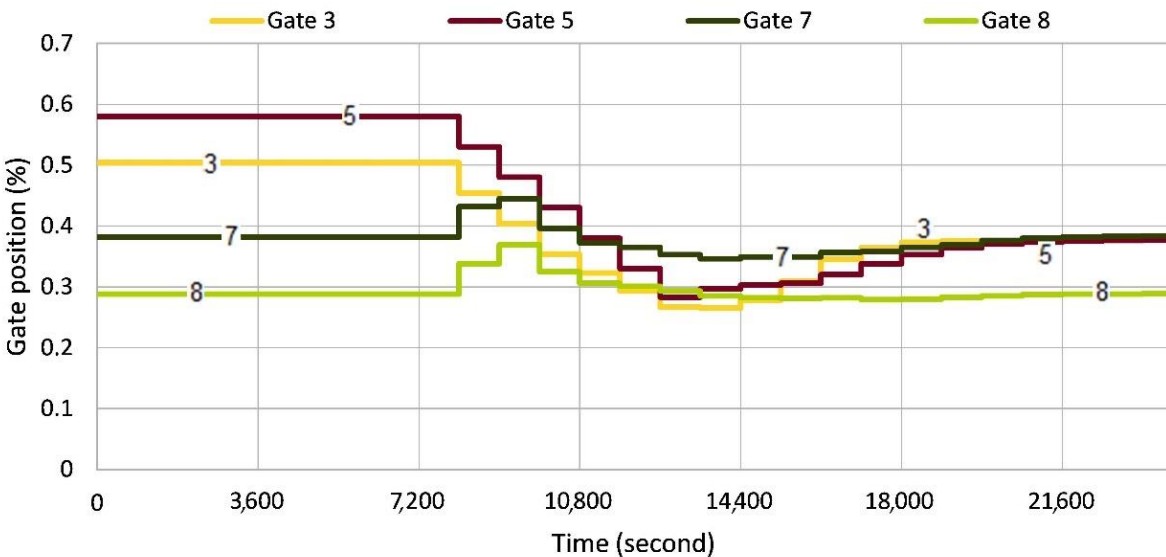

**Figure 15.** Sluicegate trajectories obtained by GoRoSoBo simplified in test 2-1 for the Corning canal (gates 3–5–7–8).

### 3.2.4. ASCE TEST CASE: Results

Once all the necessary variables and parameters that describe the Corning canal and conditions of test 2-1 have been introduced in the input of the friction coefficient algorithm, the inverse problem is ready to be solved. The initial guess value of Manning's roughness coefficients employed by the algorithm to begin the iterative optimization process was set equal to $n = 0.035$ for all Manning's trajectory parameters of the eight pools, therefore, initially assuming constant behavior of the roughness throughout the past-time horizon. The simulation of this ASCE test case took considerably longer time to execute than any of the previous practical test cases performed beforehand, which is completely expected as the number of pools, structures, and trajectory parameters implied in this test case is far greater.

The results of the Manning's roughness coefficient trajectories predicted by the FC algorithm are displayed in Figure 16 for each pool in the Corning canal. From the estimation results shown, it is deduced that in general terms, and considering that the Manning's roughness coefficient objective is to be equal to $n_{obj} = 0.020$ for all pools during the 12 h simulation, the FC algorithm successfully achieves greats results for the eight pools in the Corning canal. The average value of the computed Manning's trajectory parameters for each pool is equal to $n = 0.020$ when rounded to three significant digits, thus, proving the estimations performed actually accomplish the objective roughness values of the Corning canal. In that sense, the biggest Manning coefficient deviation is around $1.1 \times 10^{-3}$ and the average Manning coefficient is 0.02045 (that is a Manning coefficient deviation of $4.5 \times 10^{-4}$). In addition, the order of magnitude of these deviations is in line with the ones obtained in the previous example. As we have mentioned before, the FC algorithm estimates the Manning's roughness coefficient with high accuracy, because an average Manning coefficient deviation of $4.5 \times 10^{-4}$ represents water level errors less than 3.5 mm.

With regard to the Manning's coefficient estimation differences between pools, it is recognized that the estimated Manning's trajectories for some pools stand out from the rest in terms of goodness of their solution. For instance, the first pool exhibits the most accurate estimated Manning's trajectory while the second and third pools represent the opposite.

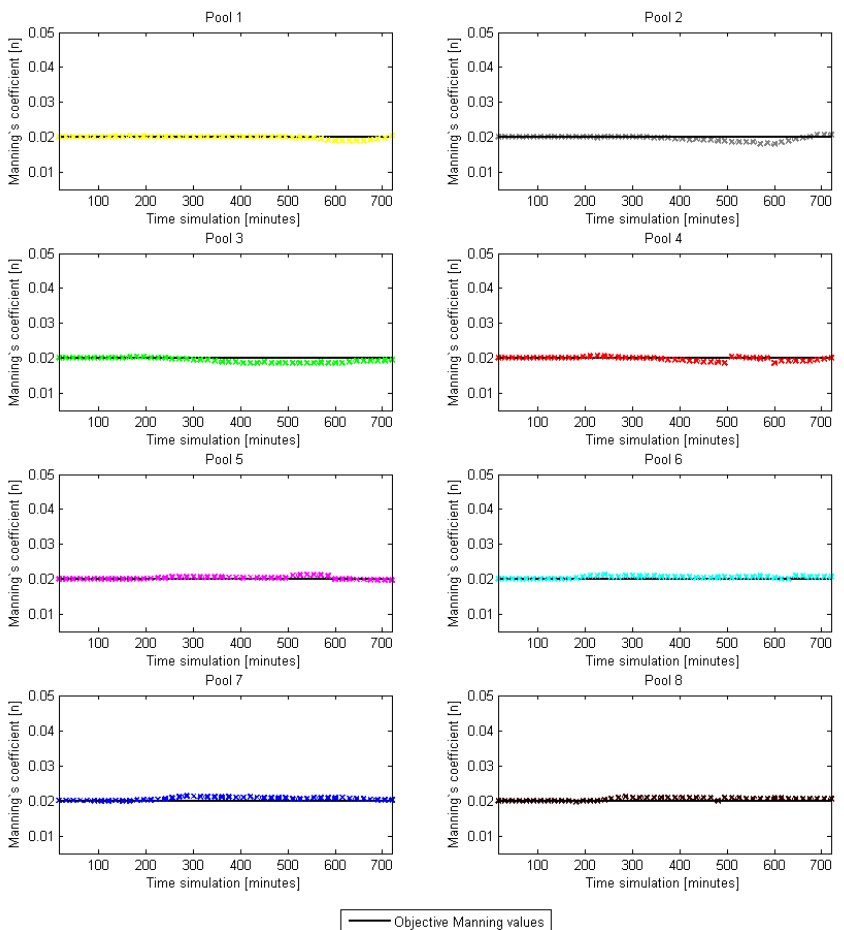

**Figure 16.** Manning's roughness coefficient trajectories obtained by FC algorithm in test 2-1 for the Corning canal (every color in the graph represent the Manning roughness coefficient calculated at every pool).

It is also interesting to note how the initial part of the predicted Manning's trajectory is consistently accurately estimated for every pool until a time instant where the predicted values start to present slight errors. This time instant is different for every pool and corresponds to the time instant when each pool starts noticing the perturbation wave generated by the introduction of unknown offtake change that propagates along the canal. This reasoning is in congruence with the time at which Manning roughness coefficient deviations appear in the estimation of the trajectories for each pool: for instance, the Manning's roughness coefficient deviations appear in pool VI around minute 180 (2 h into the simulation), which is precisely when the unscheduled offtake change occurs.

Tending to the performance indicator that compares the estimation of Manning's n trajectories with the objective values, it becomes evident that its value for this test case, $\varepsilon(n) = 1.286 \times 10^{-2}$, is not as promising as the performance indicator values from the previous practical examples. This is because the value of performance indicator ($\varepsilon(n)$) becomes larger with an increasing number of parameters, even if the differences between computed and objective values are the same for all parameters. In this sense, weighing in the magnitude factor of the problem analyzed, primarily, the 384 Manning's trajectory parameters to be identified in this test case (a canal with eight pools) in contrast with the 96 parameters identified in the canal with two pools, although the biggest Manning coefficient deviation and the average Manning coefficient have similar results in both practical examples.

All scenarios in this practical example show good results, so this is also a test of robustness of the algorithm.

## 4. Conclusions

The Manning's roughness coefficient is a highly sensitive variable in the flow behavior of any canal. In that sense, several authors have published different methodologies in order to calibrate the Manning' coefficient [10–12], which is actually the research gap.

The Manning's roughness coefficient constitutes an essential physical parameter in the behavior of open canals. The poor estimation of the value of this roughness coefficient may induce important errors in the operations established by predictive control systems in irrigation canals. This was the reason to develop the friction coefficient algorithm, which precisely aims to address this issue by developing a numerical method for estimating Manning's roughness coefficients of irrigation canals. The FC algorithm is formulated as a traditional inverse problem solver, in which the unknown parameters of the model (that is, Manning's roughness coefficients) are identified on the basis of comparing the deviations between the water levels computed by the model and a set of water-level measurements. The FC algorithm makes use of the hydraulic influence matrix (HIM), which is a matrix that establishes the influence of the Manning's roughness coefficients on the changes in the hydrodynamic state of the canal (water level and flow velocity) at all points of the canal during a past-time horizon. In that regard, the HIM is the matrix to be inverted, solving the optimization problem in order to calculate the Manning's roughness coefficient trajectories.

The optimization problem proposed by the FC algorithm is solved by means of the Levenberg–Marquardt method, which specializes in non-linear least-square problems such as the one in the FC algorithm.

The FC algorithm is tested under a wide variety of scenarios and circumstances including changes in the canal control variables, such as gate trajectories, scheduled deliveries, and pump flow trajectories. The first example is conducted in a canal with two pools with several scenarios such as changing gate position and scheduled water demands during the time horizon. In these scenarios, the FC algorithm was able to obtain a Manning roughness coefficient trajectory close to the objective Manning roughness coefficient trajectory. Regarding scenario one, the biggest Manning coefficient deviation (among real and calculated) is around $1 \times 10^{-3}$, the average Manning coefficient is 0.02498 (that is, a Manning coefficient deviation of $2 \times 10^{-4}$), and there is a performance indicator value ($\varepsilon(n)$) equal to $4.69 \times 10^{3}$. Regarding scenario two, the biggest Manning coefficient deviation is lower than $1 \times 10^{-3}$, the average Manning coefficient for the second canal pool is lower than $1 \times 10^{-4}$, and the performance indicator ($\varepsilon(n)$) is equal to $2.691 \times 10^{3}$. In all these scenarios, the algorithm accuracy is quite high because a Manning coefficient deviation of $2 \times 10^{-4}$ represents a water level deviation of less than 2 mm.

A canal simulation was performed in an adapted version of one of the test cases proposed by the ASCE task committee on canal automation algorithms, that is, test case 2-1 [17], which considers a portion of a real irrigation canal formed by eight pools. Regarding this test case, the biggest Manning coefficient deviation is around $1.1 \times 10^{-3}$, the average Manning coefficient is 0.02045 (that is, a Manning coefficient deviation of $4.5 \times 10^{-4}$), and there is a performance indicator value ($\varepsilon(n)$) equal to $1.286 \times 10^{-2}$, thus, demonstrating the robustness of the algorithm developed. Tending to the performance indicator that compares the estimation of Manning's n trajectories with the objective values, it becomes evident that its value for this test case, $\varepsilon(n) = 1.286 \times 10^{-2}$, is not as promising as the performance indicator values from the previous practical examples. This is because the value of performance indicator ($\varepsilon(n)$) becomes larger with an increasing number of parameters, even if the differences between computed and objective values are the same for all parameters. In this sense, weighing in the magnitude factor of the problem analyzed, primarily the 384 Manning's trajectory parameters to be identified in this test case (a canal with eight pools) in contrast with the 96 parameters identified in the canal with two pools (so it is not possible to compare performance indicators of both practical examples), although the biggest Manning coefficient deviation and the average Manning coefficient have similar results in both practical examples. In this scenario, the algorithm accuracy is quite high because a Manning coefficient deviation of $4.5 \times 10^{-4}$ represents a water level deviation of

less than 3.2 mm. On the other hand, the FC algorithm shows high robustness values due to the fact that the FC algorithm presents low Manning's roughness coefficient deviations in several cases and scenarios.

In light of the results obtained, it is concluded that the friction coefficient algorithm has proven to be a competent and robust algorithm for the estimation of Manning's roughness coefficients. Subsequently, its implementation in the control scheme of irrigation canals (overall control diagram) as part of the online and offline computation should be regarded. In that sense, the FC algorithm could be applied in a real irrigation canal, as the FC algorithm does not have a direct interaction with the canal actuators (which reduces the implementation issues with canal controllers) as it only estimates the changes in Manning's roughness coefficient.

**Author Contributions:** E.B.: Writing—original draft, Software, Investigation, Formal analysis; B.R.: Supervision, Resources, Funding acquisition; R.G.: Writing—original draft, Visualization, Investigation, Formal analysis; M.T.Y.: Writing—original draft, Visualization, Formal analysis; M.G.: Supervision, Writing; M.S.-J.: Supervision, Writing—review & editing, Validation, Resources. All authors have read and agreed to the published version of the manuscript.

**Funding:** This research received no external funding.

**Institutional Review Board Statement:** It is not relevant in this case.

**Informed Consent Statement:** No human test was involved in this manuscript.

**Data Availability Statement:** All data can be found tin the different papers and MD that we mentioned in the manuscript.

**Conflicts of Interest:** The authors declare no conflict of interest.

## Abbreviations

| | |
|---|---|
| $A(y)$ | The area of the wet section which depends on the water level "$y$"; |
| $c$ | The gate width; |
| $b$ | Vector of dimension $(2 \times n_S)$, $b\left[x^{k+1}, q(K)\right]$ obtained defining the $HIM(Q)$; |
| $b_j$ | The j-gate width; |
| $c(y)$ | Water wave celerity which is dependent on the water level "$y$"; |
| $C_c$ | Contraction coefficient of the gate; |
| $C_d$ | Discharge coefficient of the gate; |
| $c_T$ | Local losses coefficient in the canal; |
| $C_w$ | Discharge coefficient of a weir; |
| $d_j$ | Drop at j-gate; |
| $f(k)$ | Input function at time step $k$; |
| $g$ | Gravity; |
| $HIM(n)$ | Hydraulic influence matrix (derivative parameter n); |
| $H_{up}$ | Constant upstream water level at upstream boundary condition (canal header); |
| $I$ | Identity matrix; |
| $J(n)$ | Performance criterion (Manning coefficient trajectory) (objective function); |
| $J$ | Matrix obtained deriving the output water level vector by a variable; |
| $k_F$ | Number of sections in which the prediction horizon is divided depending on the CFL condition/time frequency of measuring data; |
| $K_F$ | Number of operation periods defined by the watermaster in which the simulating horizon is divided; |
| $k_o(k)$ | Orifice coefficient depending on its overture of the gravity i-offtake at time instant k; |
| $M$ | Matrix obtained from the Saint-Venant equation to represent the influence between parameters at different sections (in a structured mesh); |
| $N$ | Matrix obtained from the Saint-Venant equation to represent the influence between parameters at points P and Q; |
| $n$ | Manning coefficient; |

| $n_c$ | Number of checkpoints; |
|---|---|
| $n_p$ | Number of pools; |
| $n_s$ | Number of sections in which the canal is discretized; |
| $n_x$ | Dimension of prediction vector ($n_X = (2 \times n_S) \times k_F$); |
| $n_y$ | Dimension of prediction output vector ($n_Y = k_F \times n_C$); |
| $P(y)$ | The wet perimeter which is function of the water level "$y$"; |
| $Q$ | Weighting matrix; |
| $q_{offtake}$ | Discharge through the orifice offtake; |
| $q_s$ | Discharge through the lateral spillway; |
| $R_H$ | The hydraulic radius is a measure of a channel flow efficiency; |
| $r_{k_1}^{k_F}$ | Residual vector between the desired water level and the computed water level; |
| $S$ | Matrix obtained from the Saint-Venant equation to represent the influence between parameters; |
| $S(y)$ | Horizontal surface of the reception area in the checkpoint; |
| $S_0$ | Bottom slope; |
| $S_f$ | Friction slope; |
| $T^K$ | Operation instant when the Manning coefficient could be changed; |
| $T(y)$ | The maximum width dependent on the water level "$y$"; |
| $n_{ij}(K)$ | Manning coefficient of the pool i during the operation period K at the prediction horizon j; |
| $n$ | Manning roughness coefficient value; |
| $n_0$ | Vector with the Manning coefficient at the first regulation period for the previous predictive horizon; |
| $u$ | The open gate height |
| $v_i(k)$ | Mean velocity at time instant k at canal section i; |
| $x(k)$ | State vector at time instant k; |
| $y(k)$ | Subset of water depths of the state vector at time instant k at checkpoints; |
| $\mathbf{Y}_1^{k_F}$ | Predictive output vector from 1 to $k_F$; |
| $Y^*(k)$ | Subset of measurement water levels at checkpoints at time instant $k$; |
| $y_{dw}$ | Downstream j-gate water depth; |
| $y_i(k)$ | Water depth at time instant $k$ at canal section $i$; |
| $y_o$ | Height of the center of the orifice from bottom of the gravity offtake; |
| $y_s$ | Downstream water level of the reservoir; |
| $y_{up}$ | Upstream j-gate water depth; |
| $z$ | Variable used to interpolate values with the Lagrange factors; |
| $\Delta n$ | Perturbed input/output Manning vector; |
| $\Delta t$ | Numerical discretization time according to CFL condition; |
| $\Delta T$ | Operation period defined by the watermaster; |
| $\Delta Y$ | Water level perturbation; |
| $\Delta x$ | Numerical discretization space cell length; |
| $\Delta X$ | Perturbed state vector obtained when a perturbation is introduced into the system; |
| $\varepsilon(n)$ | Norm vector error between the computed and desired Manning coefficient; |

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
