# Peer review of "The FC Algorithm to Estimate the Manning’s Roughness Coefficients of Irrigation Canals"

_agriculture, doi:10.3390/agriculture13071351_

Round 1

Reviewer 1 Report

The authors develop a Friction Coefficient Algorithm. However, several issues need to be clarified. 

The research gap should be highlighted, especially when comparing with the previous similar studies conducted by the authors' group. 

In the abstract, there should be a brief description of the obtained results. 

Please check the citation style, such as in Introduction. The authors wrote that "Wahlin et al. (2006) [11].'

The majority of the references were published more than five years ago. Please add more updated references. Alternatively, please justify why you are using the old references.

Author Response

- The research gap should be highlighted, especially when comparing with the previous similar studies conducted by the authors' group. 

Ok, I will do

- In the abstract, there should be a brief description of the obtained results. 

Yes, you are right. I introduced the next sentence in the abstract:

 The FCA algorithm has been applied in several irrigation canals and different scenarios with accurate results obtaining an average Manning coefficient deviation among 2×10-4 and 4.5×10-4.

- Please check the citation style, such as in Introduction. The authors wrote that "Wahlin et al. (2006) [11].'

Ok, I do it.

- The majority of the references were published more than five years ago. Please add more updated references. Alternatively, please justify why you are using the old references.

Yes, you are right. I introduced several papers which authors provided different methodologies to calibrate Manning’s roughness coefficient.

[10] Shan-e-hyder S., Caihong H., Muhammad M. B. , Mairaj H. , 2021. Estimation of Manning’s Roughness Coefficient Through Calibration Using HEC-RAS Model: A Case Study of Rohri Canal, Pakistan. American Journal of Civil Engineering Vol. 9, Iss: 1, pp 1. DOI: 10.11648/j.ajce.20210901.11

[11] Yujian L., Yixin G., Liang M., 2020. Calibration method for Manning's roughness coefficient for a river flume model. Journal of Water Supply (2020) 20 (8): 3597-3603. https://doi.org/10.2166/ws.2020.235

[12] Yao, L., Peng, Y., Yu, X. et al. Optimal Inversion of Manning’s Roughness in Unsteady Open Flow Simulations Using Adaptive Parallel Genetic Algorithm. Water Resour Manage 37, 879–897 (2023). https://doi.org/10.1007/s11269-022-03411-x

The new version manuscript is also uploaded

Reviewer 2 Report

1.      In the setting of limited freshwater resources, how can incorporating technical breakthroughs and agricultural automation systems increase water usage efficiency?

2.      What other elements, outside the effectiveness of management systems, are the primary determinants of water use efficiency in irrigation canals?

3.      What are the current issues and restrictions with the irrigation canal management control algorithms?

4.      How are the accuracy and effectiveness of water delivery services in irrigation canals affected by the estimation of Manning's roughness coefficient?

5.      What potential effects could an inaccurate calculation of Manning's roughness coefficient have on the overall effectiveness of irrigation canal systems?

6.      How does the Friction Coefficient Algorithm (FCA) accurately calculate Manning's roughness coefficient in real-time for irrigation canals?

7.      What data inputs and sensors are required to effectively implement the Friction Coefficient Algorithm (FCA)?

8.      How does the Friction Coefficient Algorithm (FCA) handle variations in flow conditions and hydraulic parameters in an irrigation canal?

9.      Can the Friction Coefficient Algorithm (FCA) account for changes in canal roughness due to sedimentation, vegetation growth, or other factors?

10.  What is the Friction Coefficient Algorithm (FCA) accuracy and reliability compared to traditional field measurements and manual estimation methods?

11.  How can the Friction Coefficient Algorithm (FCA) be integrated with existing canal monitoring systems and data collection infrastructure?

12.  How does the Friction Coefficient Algorithm (FCA) handle uncertainties and errors in sensor measurements or input data?

13.  Can the Friction Coefficient Algorithm (FCA) adapt and dynamically update Manning's roughness coefficient based on real-time flow and channel conditions?

14.  How does the Friction Coefficient Algorithm (FCA) perform in different irrigation canals, such as concrete-lined canals, earthen canals, or canals with varying cross-sectional shapes?

15.  Can the Friction Coefficient Algorithm (FCA) be used to optimize water management and improve irrigation efficiency in canals by providing accurate and timely roughness coefficient values?

Moderate editing of English language

Author Response

Second Reviewer

Point 1.      In the setting of limited freshwater resources, how can incorporating technical breakthroughs and agricultural automation systems increase water usage efficiency?

It is possible because automation systems help you to transport the correct flowrate to every canal diversion at the correct time, so automation systems try to avoid spill water in case of water level overpass the maximum water surface level in the canal, and transport enough water in case of increasing in crops water necessities.

Point 2.      What other elements, outside the effectiveness of management systems, are the primary determinants of water use efficiency in irrigation canals?

The moisture soil sensors to identify crop water necessities is also essential in this task.

Point 3.      What are the current issues and restrictions with the irrigation canal management control algorithms?

The control algorithms deploy and implementation in a real irrigation canal is a challenge for itself as well as increasing the control algorithm reaction and be sure all communications among control algorithms, actuators and sensors deployed in the irrigation canal work properly.

Point 4.      How are the accuracy and effectiveness of water delivery services in irrigation canals affected by the estimation of Manning's roughness coefficient?

They are highly sensitive as I introduced in my thesis (Bonet E. 2015 Experimental Design and Verification of a Centralize Controller for Irrigation Canals. PhD thesis, UPC, Technical University of Catalonia, Spain), some experiments were done in a laboratory canal (CANAL-PAC-UPC) and deviations of 20% in Manning’s roughness coefficients introduced flowrate deviations of 40% in the laboratory canal (among the real flowrate and the simulating flowrate considering a Manning’s roughness coefficient deviation).

Point 5.      What potential effects could an inaccurate calculation of Manning's roughness coefficient have on the overall effectiveness of irrigation canal systems?

Errors in Canal roughness coefficients introduce a huge impact in the canal behavior as I introduced in the question before.

Point 6.      How does the Friction Coefficient Algorithm (FCA) accurately calculate Manning's roughness coefficient in real-time for irrigation canals?

The idea is quite similar than using the FCA just for planification in an off-line mode. The water level data obtained from sensors deployed in cross section in the irrigation canal is sent to the FCA algorithm which calculate the Manning’s roughness coefficients. The key is that FCA does not need too much time to estimate the coefficients just some seconds.

Point 7.      What data inputs and sensors are required to effectively implement the Friction Coefficient Algorithm (FCA)?

FCA only needs water level sensors in several cross section of the irrigation canals (one for every pool). On the other side, FCA needs geometrical canal data (slope of canal pools, geometry of cross sections, boundary flow conditions, gates positions and an initial value of Manning’s roughness coefficient (just to start with coefficient optimization))

Point 8.      How does the Friction Coefficient Algorithm (FCA) handle variations in flow conditions and hydraulic parameters in an irrigation canal?

FCA considers all conditions introduced in the input file, so HIM matrix built by the algorithm introduces all particular conditions for every time step for all of the simulating period of time, because the HIM matrix approach the influence of Manning’s roughness coefficient in flow behavior, particularly in velocity and water level in every canal cross section, so any change in Manning’s roughness coefficients is translated in the hydrodynamic conditions in every cross section in the canal for every time step.

Point 9.      Can the Friction Coefficient Algorithm (FCA) account for changes in canal roughness due to sedimentation, vegetation growth, or other factors?

For instance, sedimentation and vegetation growth introduce changes in the hydrodynamic canal conditions (for instance changes in water level in the irrigation canals), so for irrigation canal changes in water level for a particular boundary condition, the FCA estimates the Manning’s roughness coefficients changes which reproduce the current flow behavior of the canal.

Point 10.  What is the Friction Coefficient Algorithm (FCA) accuracy and reliability compared to traditional field measurements and manual estimation methods?

The main objective of FCA is to estimate the Manning Roughness coefficient and this coefficient is frequently estimated from water level measurements in the canal and calibrating the simulating models from several estimation of Manning’s roughness coefficient from iterative process.

Point 11.  How can the Friction Coefficient Algorithm (FCA) be integrated with existing canal monitoring systems and data collection infrastructure?

The common process for using the FCA algorithm is at the very beginning of the irrigation cycle, the sensors collect data from the canal, concretely water level at several cross sections of the canal, and all these data, canal geometry data and the boundary flow conditions are share with the FCA algorithm in order to estimate the Manning’s roughness coefficient.

Point 12.  How does the Friction Coefficient Algorithm (FCA) handle uncertainties and errors in sensor measurements or input data?

The errors in water levels involve errors in Manning’s roughness coefficient estimations so this is true although it would be possible to filtering the sensors measurements data in order to reduce uncertainties.

Point 13.  Can the Friction Coefficient Algorithm (FCA) adapt and dynamically update Manning's roughness coefficient based on real-time flow and channel conditions?

It would be possible, but just in case, that water level perturbations are due to Manning’s roughness coefficient deviations.

Point 14.  How does the Friction Coefficient Algorithm (FCA) perform in different irrigation canals, such as concrete-lined canals, earthen canals, or canals with varying cross-sectional shapes?

 There is not any problem, in case of Corning canal scenarios (manuscript), cross-sectional shapes change depending on the pool and the FCA worked properly.

Point 15.  Can the Friction Coefficient Algorithm (FCA) be used to optimize water management and improve irrigation efficiency in canals by providing accurate and timely roughness coefficient values?

Yes, FCA can do it considering good communications among sensors and FCA algorithm as well as the canal information (cross-section shapes, flow boundary conditions, …) is good enough.

The new version manuscript is uploaded

Reviewer 3 Report

The paper has been written with very good quality. The issue is very important to solve. I recommend publishing it. But I have one suggestion. Is there any effect on the soil layers after 20 sm?  Because usually, the root zone of tomato crops is about 80 sm. But in this article, soil deep taken into account only 20sm.

Author Response

Third reviewer

The paper has been written with very good quality. The issue is very important to solve. I recommend publishing it. But I have one suggestion. Is there any effect on the soil layers after 20 sm?  Because usually, the root zone of tomato crops is about 80 sm. But in this article, soil deep taken into account only 20sm.

Yes, you are right. Although the crop water demands (tomato water demands) is input data for the FCA algorithm, so FCA is not worry about this information, because the moisture soil estimations as well as the crop water necessities are known variables in this process and are out of scope of this manuscript.

Reviewer 4 Report

The paper ”The FCA algorithm to estimate the Manning’s roughness coefficients of irrigation canals” (authors: E. Bonet, R. González, M. Gómez, M. T. Yubero & M. Sánchez-Juny) has been reviewed.

There are some suggestions and comments:

1. It is better not to use abbreviations (FCA – 26 l.) in the abstract of the paper. It was mentioned again in line 84.

2. Please, choose: “Friction Coefficient Algorithm (FC algorithm)” or “Friction Coefficient Algorithm (FCA)”.

3. It is better to write “[6, 7]” instead of “[6] [7]”.

4. Please, number the chapters (for example: “2. Materials and methods”) in the case of the usage of subchapters 2.1 and 2.2.

5. The quality of Figure 3 should be increased.

6. The description of Figure 3 must be expanded (a …, b…, c… and d…).

7. Number 2.1 was used twice for the subchapters.

8. Research methodology should be described more clearly and comprehensibly to the reader. Was the research performed as an analytical or as a numerical simulation? Which software was used?

9. The experiment is described in the results chapter. Maybe it is better to move the description of the methodology to the second chapter.

10. Was it a numerical experiment (chapter 3.1)? What is the meaning of “numerical example”?

11. The model accuracy and the results validation questions must be discussed in the paper.

12. The authors could devote one conclusion to the discussion of the applicability of the research results in practice.

13. Please, correct or delete the sections: “Supplementary Materials”, “Author Contributions”, and so on.

14. In conclusion: The methodology should be described more clearly. Separate results from methodology. The question of the model validation and the results’ practical use should also be discussed in more detail.

Conclusion

The results are original and interesting, and I think that this paper may be published in the journal but only after major revisions.

Author Response

Fourth reviewer

There are some suggestions and comments:

Point 1. It is better not to use abbreviations (FCA – 26 l.) in the abstract of the paper. It was mentioned again in line 84.

Ok, I removed the abbreviation in the abstract.

Point 2. Please, choose: “Friction Coefficient Algorithm (FC algorithm)” or “Friction Coefficient Algorithm (FCA)”.

Yes, It is true. We choose “Friction Coefficient Algorithm (FC algorithm)”

Point 3. It is better to write “[6, 7]” instead of “[6] [7]”.

Ok, it has been modified.

Point 4. Please, number the chapters (for example: “2. Materials and methods”) in the case of the usage of subchapters 2.1 and 2.2.

Ok, it has been modified.

Point 5. The quality of Figure 3 should be increased.

Ok, it has been improved.

Point 6. The description of Figure 3 must be expanded (a …, b…, c… and d…).

Ok, it is done.

Point 7. Number 2.1 was used twice for the subchapters.

Ok, it has been solved.

Point 8. Research methodology should be described more clearly and comprehensibly to the reader. Was the research performed as an analytical or as a numerical simulation? The Saint Venant equations are solved numerically because they cannot be solved analytically, and the FC algorithm solves the hydraulic inverse problem associated to the Saint Venant equations, so all simulations solve a numerical problem. In any case, I include new citations to make more understandable the methodology for the reader. Which software was used? The software has been developed for ourselves in order to develop the algorithm.

Point 9. The experiment is described in the results chapter. Maybe it is better to move the description of the methodology to the second chapter.

At the second chapter, it has been introduced the control system scheme, the theory associated to the methodology and the optimization problem of the algorithm. On the other hand at the chapter 3, the algorithm is checked in several practical cases, introducing those practical examples and show the results. Although I totally agree with you, we have to change the name of chapter 3, because it is quite strange introduce the results without introducing the cases, so we change the name of chapter 3 to be more understandable for the reader.

Point 10. Was it a numerical experiment (chapter 3.1)? What is the meaning of “numerical example”?

Yes, probably it is not the best description. We change it by practical example.

Point 11. The model accuracy and the results validation questions must be discussed in the paper.

Yes, this is the reason why we introduce the performance indicator to evaluate the accuracy of the model and the robustness of the model is showed from the accurate results in several different scenarios. We discuss the model accuracy and robustness in every sub-section “results” for every scenario and also in the conclusion section, but in any case, we include new discussions about accuracy and robustness of the algorithm. On the other hand, the algorithm validation is done according to the practical cases results.

Point 12. The authors could devote one conclusion to the discussion of the applicability of the research results in practice.

Yes, it has been introduced

Point 13. Please, correct or delete the sections: “Supplementary Materials”, “Author Contributions”, and so on.

Ok, we have deleted these sections.

Point 14. In conclusion: The methodology should be described more clearly. Separate results from methodology. The question of the model validation and the results’ practical use should also be discussed in more detail.

Yes, I agree: I separate practical cases description VS practical cases results, so I introduced more sub-sections, and I included new discussions about the results. On the other hand, I also include more information in the methodology also including new citations.

Round 2

Reviewer 1 Report

The paper has been well revised. I have no further comment about the paper.

Reviewer 2 Report

The comments are incorporated in the revised version. 

 Minor editing of the English language required

Reviewer 4 Report

The revised version of the paper ”The FC algorithm to estimate the Manning’s roughness coefficients of irrigation canals” (authors: E. Bonet, R. González, M. Gómez, M. T. Yubero & M. Sánchez-Juny) has been reviewed.

It can be noticed that the authors revised, corrected and added additional information to the paper. The manuscript was improved.

Conclusion

The results are interesting, and I think that this paper may be published in the journal.